# Experiences on the implementation and maintenance of the Canadian Disability Participation Project: A mixed-methods study

Femke Hoekstra[1,2,3]*, Alanna Shwed[3,4], Sarah Lawrason[3,4], Kathleen A. Martin Ginis[2,3,4,5], Veronica Allan[6], Anita Kothari[7], Christopher B. McBride[8], Heather L. Gainforth[3,4]

**1** Department of Medicine, Division of Social Medicine, The University of British Columbia, Vancouver, British Columbia, Canada, **2** Centre for Chronic Disease Prevention and Management, The University of British Columbia, Kelowna, British Columbia, Canada, **3** International Collaboration on Repair Discoveries (ICORD), The University of British Columbia, Vancouver, British Columbia, Canada, **4** School of Health and Exercise Sciences, The University of British Columbia, Kelowna, British Columbia, Canada, **5** Department of Medicine, Division of Physical Medicine & Rehabilitation, The University of British Columbia, Vancouver, British Columbia, Canada, **6** Sport Information Resource Centre, Ottawa, Ontario, Canada, **7** School of Health Studies, Western University, London, Ontario, Canada, **8** Spinal Cord Injury BC, Vancouver, British Columbia, Canada

* femke.hoekstra@ubc.ca

## Abstract

Establishing a multidisciplinary network of researchers, trainees, and research users—such as the Canadian Disability Participation Project (CDPP)—is a promising approach to promote and support research partnerships and improve the application of disability research findings. This study aimed to 1) describe the implementation of the CDPP network over time and 2) explore members' experiences and reflections on the implementation and maintenance of the CDPP network and its partnerships. This mixed-methods study used survey data, collected among CDPP researchers, trainees and research users in the years 2018, 2019 and 2021, and interview data, collected at the end of the study period (2021/2022). Survey items, focused on network functioning and satisfaction (implementation), were analyzed using descriptive statistics. Interviews focused on members' experiences and reflections of the implementation and maintenance of the network and its partnerships, and were analyzed using reflexive thematic analysis. Members were positive about how the network functioned and satisfied with how the CDPP implemented its plans. Over 70% of the survey participants indicated that it was easy to work with researchers/research users in the CDPP network (2018: 71%; 2019: 85%; 2021: 70%). Interview participants discussed the strong leadership of the network, the lack of feeling meaningfully connected to the network as a whole, and key principles that guide the success of individual research partnerships *(implementation).* Participants reported that (human) resources and continued leadership are needed to sustain the network and its partnerships long-term *(maintenance).* This study provides unique longitudinal insights

**Data availability statement:** The survey data as well as other supporting documents (survey questions and interview guide) are publicly available from Open Science Framework (https://osf.io/rwf2u/).

**Funding:** This study was supported by a partnership grant from the Social Sciences and Humanities Research Council of Canada (grant no. 895-2013-1021) for the Canadian Disability Participation Project (www.cdpp.ca). The funders had no role in study design, data collection and analysis, decision to publish, or preparation of the manuscript.

**Competing interests:** The authors have declared that no competing interests exist.

into the implementation of a multidisciplinary network of research partnerships. The findings highlighted that building and sustaining a large network of partnerships is challenging and requires strong and continued leadership. To conclude, we describe lessons learned for research partnership capacity building and the translation of disability research to practice and policy.

## Introduction

Conducting and/or disseminating research in partnership with community members (e.g., policy-makers, practitioners, community organizations, patients) has been proposed as a promising and important approach to close the gap between research and practice [1–6]. While many partnerships have been established within networks [7–9] and individual projects [10–12], developing and sustaining these partnerships is complex. An in-depth understanding of how research partnerships are implemented over time is needed to improve and build capacity for meaningful partnerships and to advance the science in this domain [13–15].

One opportunity to better understand how research partnerships are implemented over time is through formally funded multidisciplinary networks of researchers and community organizations. Indeed, previous network initiatives in different domains (e.g., domestic violence [16], mental health [17], obesity prevention [18]) have reported that these networks can stimulate collaborations between researchers and community partners (i.e., research users) and contribute to improved uptake of research findings. Moreover, these networks in which researchers and research users (i.e., individuals or groups that use or benefit from the research) work in partnership on different projects, provide an ideal context to study the implementation and sustainability of these networks and their research partnerships [19].

Previous studies reporting on networks and their partnerships have shown that implementing partnership approaches is complicated due to the many barriers that researchers, trainees and research users experience [20,21]. Barriers to (disability) research partnerships are substantial time commitment, communication challenges among members of the partnership, and ambiguous roles and responsibilities [21,22]. Furthermore, additional (financial) barriers exist to effectively sustain these networks and their partnerships over the longer time. In contrast, facilitators to research partnerships are strong interpersonal relationships among members of the partnership, a shared vision, and effective communication [21,22]. Accordingly, establishing a network of researchers and research users (i.e., network of partnerships) may facilitate relationship building and support research partnership approaches. However, the majority of research examining partnerships is concentrated on members' experiences with being part of an individual partnership project. Less is known about members' experiences of being part of a large network of partnerships, in which benefits might be amplified. Furthermore, most research is cross-sectional and conducted outside the field of disability research. Establishing and sustaining partnerships with equity-owed populations, such as people with disabilities, may add additional

complexities and may require specific guidance [3,6,23,24]. Understanding how to effectively implement and sustain a network of research partnerships may contribute to improving the uptake of research findings into practice or policy.

The Canadian Disability Participation Project (CDPP) is an example of a large, multidisciplinary network of partnerships in the areas of disability and participation. The CDPP was established in 2015 to enhance the quantity (i.e., the number of people who participate) and quality (i.e., subjective experiences of participation, including satisfaction and enjoyment [25]) of community participation among Canadians with physical disabilities in three areas: employment, mobility, and sport and exercise [19]. This network consisted of >31 principal researchers and >19 community organizations who work in partnerships to conduct and/or disseminate their research. While all CDPP projects focused on enhancing participation among people with disabilities, they differed in their research scope, topic, design, and degree of research user engagement. Recognizing that the CDPP provided an ideal opportunity to study a network of partnerships, we developed a protocol to longitudinally study the implementation and sustainability of the network using quantitative (logs, surveys) and qualitative research methods (exit and timeline interviews). Data were collected from different perspectives (researchers, trainees and research users) at different time points between the years 2018 and 2022.

The overarching aim of the CDPP evaluation study was to assess and explain the impact of the CDPP network using the RE-AIM framework [26]. This framework includes five components: Reach, Effectiveness, Adoption, Implementation and Maintenance. The evaluation protocol, which is described elsewhere [19], elaborated on the work of Sweet et al. [27], who used the RE-AIM framework to evaluate the impact of a large partnership between researchers and research users. This paper reports on the implementation and maintenance components. Implementation focused on CDPP members' views on how the network functioned, their satisfaction over time and their experiences of being part of the network and its partnerships. The maintenance component focused on CDPP members' views on the future of the network and its partnerships. More specifically, this study aimed to:

1) Describe the *implementation* of the CDPP network over time;

2) Explore members' experiences and reflections on the *implementation* and *maintenance* of the CDPP network and its partnerships.

Together, these insights provide directions to improve and build capacity for research partnerships among researchers, trainees and research users working in the area participation and disability.

## Materials and methods

### Project overview

This study used a retrospective longitudinal sequential mixed methods design [28], in which we collected data at different times points between 10 August 2018 and 22 April 2022. Survey data were collected among CDPP members at three measurement points in the year 2018, 2019 and 2021, between 10 August 2018 and 4 November 2021. Due to the COVID-19 pandemic, we did not collect survey data in the year 2020. Interview data were collected at the end of the study period between 21 November 2021–22 April 2022. This paper reports on survey and interview data related to the implementation and maintenance components of the RE-AIM framework [26]. This paper includes a brief overview of the methods and procedures. Further details are available in the protocol paper [19]. Our original plan was to collect survey and interview data to explore members' experiences with being part of a network of partnerships (network-level data) and to collect additional survey and interview data to explore members' experiences with conducting and disseminating a specific research project in partnership (partnership-level data) [19]. However, we modified this plan to avoid overburden of participants. We did not collect the partnership-level survey data and combined experiences on both levels (partnership and network) into one interview session.

We conducted this study using a pragmatic approach. Pragmatism follows an ontological relativist paradigm with a specific emphasis on practical outcomes of the knowledge within a particular context [29]. Within pragmatism, subjective interpretations of experiences (i.e., intersubjectivity) are valued rather than seeking a single objective truth, making a pragmatic approach methodologically coherent with the use of mixed methods [30]. Aligning with our pragmatic approach, this project was conducted by a multidisciplinary expert team, including researchers and trainees with expertise in research partnerships, network function, implementation, knowledge translation, and/or disability and one community partner with expertise in disability, network function and partnerships. Team members were meaningfully engaged at appropriate times based on their interest and capacity [31]. S1 Table provides additional information about the team, including a positionality statement and S2 Table provides additional details on our collaborative research activities. Ethical approval for this study was obtained from the Behavioural Research Ethics Board (BREB) of the University of British Columbia – Okanagan campus (Approval Number: H18-00459). All participants provided written consent by reviewing a document that detailed the study's purpose, procedures, potential risks, and benefits. Participants in the interview study also provided oral consent prior to the interview session.

### The CDPP network

The overarching goal of the CDPP was to enhance the quantity and quality of community participation among Canadians with physical disabilities. The CDPP network was funded by a 7-year Partnership Grant from the Social Sciences and Humanities Research Council (SSHRC). The aim of Partnership Grants is to support formal partnerships to promote research, research training and/or KT in the fields of social sciences and humanities.

The CDPP included one leadership team and three teams focused on three different participation domains (mobility, employment, sport/exercise), and a fourth team focused on participation issues that cut across domains. The leadership team included the project director, executive members, content experts, and team leads. The governance structure was established with the aim of promoting open and transparent decision-making and ensuring accountability and effective communication. Additionally, deliberate choices were made in the leadership and management structure to incorporate both academic and non-academic perspectives, fostering shared leadership at all levels. These arrangements were important to ensure a balance of power and influence through shared decision-making, development of community-driven research questions, co-learning, mutual contributions of expertise, co-mentorship of trainees, and co-ownership of the research process and products. To illustrate, each domain team was co-led by an academic and a non-academic who worked together on the development and implementation of their team's research-related activities and training. Bi-monthly teleconference/ online meetings served as a platform for discussing ongoing projects, emerging issues, and shared challenges. Within this structure, each academic team leader was paired with a research coordinator. The research coordinator's responsibilities included participating in the teleconferences/meetings, disseminating relevant information (such as newsletters, meeting minutes, and bulletins) to team members via email, and tracking budget allocations and KT efforts for reporting purposes.

### Study population

The study population included researchers, trainees and research users (i.e., CDPP members) affiliated with the CDPP network. In the context of CDPP, research users included representatives of community organizations, people with disabilities and their key social influencers (coaches, caregivers), employers, and policy-makers. CDPP members were invited to complete a 10-minute online survey in the years 2018, 2019 and 2021 by a member of the research team (FH, KMG). In 2019, we asked CDPP researchers to forward the survey invitations to their partners (research users). Using purposeful sampling [32], a selection of CDPP members (researchers, trainees, and research users) were invited to participate in an online semi-structured interview session at the end of the study period (2021/2022). We selected potential participants using their 2021 survey responses and their position in the network based on their research domain (employment, mobility

or sport and exercise) and their role (researcher, trainee, research user). We invited CDPP members who indicated that they were willing to participate in an interview session about their CDPP experiences.

## Data collection

**Surveys.** CDPP members were invited to complete an annual survey about their views on how the network functioned (i.e., implementation) and their satisfaction levels of being part of the CDPP network. Network functioning was assessed using four self-constructed items focusing on the extent to which the goal and value of the CDPP were clear (2 items), and how easy and useful it was to work together with community partners/researchers within the CDPP (2 items). Network satisfaction was assessed with five self-constructed items focusing on participants' views and satisfaction about their role and influence within the CDPP network. Changes in participants' views and satisfaction levels were assessed using survey items in all three years (2018, 2019, 2021).

**Interviews.** Interviews were conducted to explore members' experiences of the implementation and maintenance of the CDPP network and their experiences with conducting and disseminating CDPP research in partnership. The first author (FH) drafted the first version of the interview guide in collaboration with co-authors with expertise in qualitative research, partnership research and/or network functioning (VA, AL, HG, SL, AK). Interview questions were created by building on questions from a previous interview study on researchers' and research users' experiences with conducting and disseminating spinal cord injury research in partnership [33]. Minor editorial changes to the interview guide were made after conducting the first interview session. The interview guide included questions about members' experiences with the CDPP network, their perceived successes and challenges, their view on the future of the CDPP and their partnership experiences. Example questions were: "*Tell me about your experiences being part of the CDPP network*." and "*If any, what are, in your opinion, some of the successes of the CDPP network?*". Members' views on the future of the CDPP were explored during the interview sessions. An example interview question was: "*In your opinion, how should CDPP network function after 5 years?*". Additional follow-up questions were asked to provide participants opportunities to explain and elaborate on their responses. The full interview guide is available in S1 Text. The interviews were conducted online by SL, FH or VA using UBC's Zoom video software methods. Multiple interviewers were required to recognize power differences between interviewers and participants. Namely, the interviewer and participant could not have a mentor-trainee relationship with each other.

## Data analyses

Survey data were analyzed using descriptive statistics to identify changes in network functioning and satisfaction levels over time. Results of the survey items were reported using medians, ranges, means and standard-deviation. As each year, the group of participants differed due to staff turnovers and graduation of trainees, we did not test for changes over time.

Interviews were audio recorded, transcribed verbatim by a research assistant, and checked for accuracy by the interviewer (FH, SL). Names in the transcripts were anonymized. A reflexive thematic analysis approach was taken to analyze the interview data [32,34], using NVivo software. First, co-author (AS) became familiar with the interview data, reviewed the transcripts to identify codes and then constructed initial themes. The coding and construction of initial themes were done by co-author (AS) who did not conduct any of the interviews to minimize the risk that experiences from certain participants would get more attention as interview sessions were conducted by different team members. Second, four co-authors (FH, SL, AS, HG) met to discuss and clarify those initial themes over a 1-hour in-person meeting. After the meeting, AS then further refined, defined, and named a next iteration of themes. The same four co-authors met again for another 1-hour meeting to discuss the refined themes. A draft of refined themes and sample quotes were shared with other co-authors (KMG, CBM, AK, VA) who acted as "critical friends" (i.e., a trusted person who asks challenging questions, another lens, and offers critiques) [35]. Based on their feedback, themes and descriptions were further refined and incorporated into the manuscript. Lastly, the first author (FH) finalized theme names and descriptions based on multiple

rounds of feedback from all co-authors. S2 Table provides further details on how themes were developed based on our collaborative research activities and input from co-authors. Quotes from participants were included to illustrate the themes and findings. Pseudonyms were used to enhance readability.

## Methodological rigour

Various steps were taken to enhance the quality and methodological rigour of this study. First, themes were constructed through discussions with co-authors with various backgrounds and lived experiences and by using 'critical friends' who reflected on data findings. Reflexivity also included a description of the research team including a positionality statement which is available in S1 Table. Second, we meaningfully engaged research users (community partner and network leaders) as partners in the research process to ensure our findings are relevant and meaningful to them (worthy topic). Aligning with our pragmatic approach, we formulated lessons learned for implementing and sustaining networks of partnership in the area of disability research. Third, we used theory and previous literature to interpret and reflect on our findings to enhance the credibility of our findings and achieve meaningful coherence.

## Results

### Survey

Table 1 presents demographics of participants who completed the annual survey in 2018, 2019 and 2021.

Table 2 provides a summary of the participants' responses on items about the CDPP network functioning and their satisfaction levels. At all time points, the majority of the participants indicated that the goal of CDPP network is clear (2018: 85%; 2019: 97%; 2021: 76%). More than 70% of the participants indicated that it is easy to work with researchers/research users in the CDPP network (2018: 71%; 2019: 85%; 2021: 70%).

**Table 1. Demographics survey responses.**

|  | Survey responses | | |
|---|---|---|---|
|  | 2018<br>n=44 | 2019<br>n=36 | 2021<br>n=33 |
| **Researchers (n)** | **16** | **15** | **18** |
| Age (mean±SD) | 42.6±8.5 | 47.8±10.7 | 46.4±8.7 |
| Gender N (%) |  |  |  |
| Women | 9 (56%) | 9 (60%) | 10 (33%) |
| Men | 7 (44%) | 5 (33%) | 5 (67%) |
| Prefer not to answer/ not listed | 0 | 1 (7%) | 0 |
| **Trainees (n)** | **22** | **11** | **9** |
| Age (mean±SD) | 31.2±8.7 | 30.9±4.5 | 30.7±8.8 |
| Gender N (%) |  |  |  |
| Women | 17 (77%) | 9 (82%) | 6 (67%) |
| Men | 5 (23%) | 2 (18%) | 3 (33%) |
| Prefer not to answer/ not listed | 0 | 0 | 0 |
| **Research users (n)** | **6** | **10** | **6** |
| Age (mean±SD) | 49.8±9.7 | 50.1±13.9 | 47.8±9.6 |
| Gender N (%) |  |  |  |
| Women | 4 (67%) | 5 (50%) | 2 (33%) |
| Men | 2 (33%) | 3 (30%) | 4 (67%) |
| Prefer not to answer/ not listed | 0 | 2 (20%) | 0 |

**Table 2. Members' views on network functioning and satisfaction (implementation).**

| CDPP network functioning items | 2018 (n = 44) | 2019 (n = 36) | 2021 (n = 33) |
|---|---|---|---|
| *Network functioning* | | | |
| The **goal** of the CDPP network is clear. | Median: 7 (Range: 4–7) 6.4 ± 0.8 *Unsure: 4.9%* | Median: 6 (Range: 3–7) 6.2 ± 0.9 *Unsure: 0%* | Median: 6 (Range: 2–7) 5.6 ± 1.5 Unsure: 0% |
| The **value** of the CDPP network is clear. | Median: 7 (Range: 3–7) 6.3 ± 0.9 *Unsure: 2.4* | Median: 7 (Range: 3–7) 6.4 ± 0.9 *Unsure: 0%* | Median: 6 (Range: 2–7) 5.7 ± 1.4 *Unsure: 0%* |
| It is **easy** to work together with community partners/researchers in the CDPP network. | Median: 5 (Range: 3–7) 5.2 ± 1.5 *Unsure: 7.3%* | Median: 6 (Range: 3–7) 5.9 ± 1.2 *Unsure: 6.9%* | Median: 5.5 (Range: 1–7) 5.3 ± 1.7 *Unsure: 9.1%* |
| It is **useful** to work together with community partners/researchers in the CDPP network. | Median: 7 (Range: 2–7) 6.5 ± 1.0 *Unsure: 2.4%* | Median: 7 (Range: 5–7) 6.6 ± 0.6 *Unsure: 3.4%* | Median: 7 (Range: 2–7) 6.3 ± 1.2 *Unsure: 3.0%* |
| *Network satisfaction* | | | |
| I am satisfied with the way the people and organizations in the CDPP network work together. | Median: 6 (Range: 3–7) 5.6 ± 1.3 *Unsure: 4.9%* | Median: 6 (Range: 3–7) 5.8 ± 1.1 *Unsure = 6.9%* | Median: 6 (Range: 2–7) 5.5 ± 1.4 *Unsure: 6.1%* |
| I am satisfied with **my influence** in the CDPP network. | Median: 6 (Range: 1–7) 5.3 ± 1.6 *Unsure: 4.9%* | Median: 6 (Range: 3–7) 5.8 ± 1.2 *Unsure: 13.8%* | Median: 5 (Range: 1–7) 5.2 ± 1.6 *Unsure: 9.1%* |
| I am satisfied with **my role** in the CDPP network. | Median: 6 (Range: 2–7) 5.6 ± 1.5 *Unsure: 4.9%* | Median: 6 (Range: 3–7) 5.9 ± 1.2 *Unsure: 6.9%* | Median: 5 (Range: 2–7) 5.3 ± 1.5 *Unsure: 6.1%* |
| I am satisfied with the CDPP's plans for achieving its goals. | Median: 6 (Range: 2–7) 5.9 ± 1.2 *Unsure: 2.4%* | Median: 6 (Range: 3–7) 5.9 ± 1.1 *Unsure: 13.8%* | Median: 6 (Range: 2–7) 5.5 ± 1.4 *Unsure: 9.1%* |
| I am satisfied with the way the CDPP implemented its plans. | Median: 6 (Range: 2–7) 5.9 ± 1.2 *Unsure: 9.8%* | Median: 6 (Range: 3–7) 5.9 ± 1.1 *Unsure: 10.3%* | Median: 6 (Range: 4–7) 5.9 ± 1.2 *Unsure: 12.1%* |

Participants' responses towards their satisfaction levels were mixed, but generally positive. To illustrate, the majority of the participants (2018: 85%; 2019: 85%; 2021: 68%) indicated that they were satisfied with the CDPP's plans for achieving its goals. Similarly, the majority of the participants (2018: 81%; 2019: 88%; 2021: 70%) indicated that they were satisfied with the way the CDPP implemented its plans. S3 Table provides the survey findings disaggregated by participants' role (researchers, trainees and research users). These disaggregated results illustrate that, on average, research users reported higher satisfaction levels regarding their influence and their role in the CDPP network at all time points compared to researchers and trainees. Research users reported, on average, higher levels of agreement compared to researchers and trainees on the statement "*It is easy to work together with community partners/researchers in the CDPP network*".

Survey items were rated on a 7-point Likert scale, in which 1 = strongly disagree, 4 = neither disagree nor agree, and 7 = strongly agree. The number of participants who responded 'unsure' were excluded from these analyses, as we

assumed that these participants were unable to comment on the item (for example due to their limited involvement or role in the network). The percentage of participants who responded 'unsure' is listed for each item. S3 Table provides aggregated findings by researchers, trainees and research users.

## Interviews

A total of 15 participants took part in the interview, of which 11 (73%) identified as women and 4 (27%) as men. The mean age was 43.6 ± 10.2 years old. Regarding the participants' roles in the CDPP, 4 (27%) participants were trainees, 4 (27%) were researchers, 4 (27%) research users and 3 (20%) participants had both a researcher and research user role in the network. The interview sessions lasted on average 53.7 ± 11.4 minutes.

Overall, interview data illustrated that participants viewed the implementation of the CDPP as successful due to its growth in connections and individual research partnerships over time. A lead researcher of the CDPP said: *"I wanted scientists to learn how to do partnered research if they weren't already doing it and then to implement it and do it. And I see that at the end of the day, we accomplished that"*. Participants shared their views on the future of the CDPP network (maintenance) and provided directions to sustain meaningful collaborations and partnerships. Table 3 provides a summary of each theme focusing on the implementation and maintenance of the CDPP.

## Implementation

**Strong leadership and mentorship are the foundation for success.** A common theme of discussion among researchers was the strong leadership team and skills of the CDPP. Leaders provided a clear and collective vision for the network, facilitated communication methods, and demonstrated partnership values among a large, multidisciplinary network. The leadership of the CDPP provided members with the skills and autonomy to build and lead their own individual partnerships. Members viewed the guiding leadership as a strength of the network, especially since three areas of research (mobility, employment, and sport & exercise) worked together:

*"I think leadership is a strength of the CDPP… It [CDPP] brought together researchers from across Canada… It [CDPP] identified three pillars of participation…"* (John, researcher)

Table 3. Overview of the identified themes from the interviews with CDPP members.

| RE-AIM component | Theme | Description |
|---|---|---|
| Implementation | *Strong leadership and mentorship are the foundation for success* | A strong leadership team was required to successfully build a large, multi-disciplinary network. Mentorship from leaders, who acted as role models, in partnerships helped trainees and members to have positive partnership experiences. |
| | *The CDPP functions as many individual partnerships rather than one united network* | While members felt meaningfully connected to their individual partnerships, they did not feel meaningfully connected to the CDPP network. Connections across individual partnership teams and domains were lacking. |
| | *Key principles guide the success of research partnerships* | CDPP members identified important principles for their research partnerships: 1. Partners recognize the value of research partnership 2. Partners mutually benefit when working in research partnership 3. Human connection is the foundation for meaningful and respectful partnership 4. Partners foster honest and transparent communication |
| Maintenance | *Build capacity to sustain the network and its partnerships* | To sustain a large network and its partnerships, capacity is needed for successful leadership turnover and sustained mentorship opportunities. Effective capacity building among researchers, research users and trainees requires resources, including: 1) Human resources 2) Money and time 3) Training |

KT = knowledge translation.

Researchers recognized the value of uniting three areas and acknowledged that building a nationwide network, like the CDPP, is not easy. One researcher mentioned that they underestimated how difficult it would be to bring together researchers from different areas:

"*I think I underestimated how difficult it would be to bring those three streams together…the challenges were that I was working with three different streams…we're working in three very different scientific contexts…I think ultimately the challenges were around different phases of science all within the SSHRC spectrum and weaving those together. I think another big challenge was that I had three streams whose partnerships were at different points when we started the project…there were certain things that we could really get going in one team, and at the other team we were still doing some fundamental partnership building.*" (Mady, researcher)

Researchers and trainees also discussed that the mentorship within this network was incredibly helpful and effective for developing skills and for advancing careers. Researchers said that mentoring provided by CDPP leads, who acted as role models, helped to facilitate the growth of potential future leaders within the network. One researcher explained how another researcher was involved as a member in the network and they transformed to more of a leadership role as time went on.

**The CDPP functions as many individual partnerships rather than one united network.** Although participants perceived the growth of the CDPP as impressive and helpful for advancing collaborations and partnerships, participants mentioned it was difficult to foster cohesion among a network of this size. One researcher explained that the sheer size of the network makes it difficult to unite the CDPP as one group:

"*The size of it is a strength, but I'd say it's also a limitation…I think that it is hard if the CDPP operates as one and you have these three areas, I didn't know what was going on with the other two…I don't know if I've ever seen the whole group come together.*" (Emma, researcher)

Across interviews, CDPP members indicated that they did not feel meaningfully connected to the CDPP network. Even with these individual partnerships and obvious research partnership skills, connection among partnerships and across the research areas was perceived as lacking. A community research user explained that communication through a large entity like the CDPP is challenging:

"*I just think the communication is always the hardest thing to do in a network…and I think there has been challenges in terms of staying connected to the project sometimes, and connected to the project as a whole, because we're connected through different projects within the whole CDPP.*" (Oliver, research user)

Participants mentioned that the CDPP functioned as individual teams or partnerships, not as one united group or network. Some researchers and research users wanted more engagement across the network with other partnerships moving forward. One research user explained:

"*I think we have a really unique opportunity too, we're at so many different tables that we can really have an opportunity to unite and ensure that there's not overlap and rather complimentary efforts, but we just need to figure out a mechanism that's sufficient to do that.*" (Sophia, researcher and research user)

**Key principles guide the success of research partnerships.** Across all interviews, participants talked about their experiences with conducting and disseminating CDPP research in partnership. Participants talked about principles or values that they think are important to guide a successful research partnership. These key partnership principles

related to (1) valuing research partnerships, (2) mutual benefit, (3) human connection, and (4) honest and transparent communication.

1) Partners recognize the value of a research partnership approach

Participants emphasized the importance of recognizing the value a research partnership approach and diverse forms of expertise for ensuring a successful partnership. They explained that research partnership approaches should not be adopted just to meet a requirement, but the drive to want to engage with research users must be present. One trainee explained that for meaningful partnerships, those involved need to value partnered over traditional, unpartnered research approaches. Another CDPP member explained that if researchers do not value a research partnership approach then both the partnership and future partnership work will suffer:

*"So you got to really want to do it [research partnership]. Like don't go into this just to tick a box. [..] You have to want to do this, because if you're not genuine, it's going to fall flat, you're going to do a disservice."* (Mady, researcher)

2) Partners mutually benefit when working in research partnership

Participants emphasized that in a successful partnership, everyone should feel they are gaining from being part of the partnership (e.g., new and useable knowledge). Mutual benefit was identified by both researchers and research users as a key to meaningful partnership. They explained that mutual benefit including working following a strength-based approach to ensure that both groups benefit. One research user described mutual benefit as follows:

*"I think that all partners should feel like they're both gaining something and contributing something, that they're able to get something that helps them better their work and that they're able to offer something that contributes to the partnership and makes it more effective."* (Ava, research user)

3) Human connection is the foundation for meaningful and respectful partnership

Across all discussions about partnership principles, human connection (i.e., relationships between partners) was identified as the foundation for meaningful and respectful partnership. Participants highlighted that respect for each partner's expertise, time, and ability to contribute is a key feature in their successful partnerships. One researcher explained that listening to and valuing what each partner brings to the project is important for fostering those relationships:

*"Listen to your partners…appreciate where your partners are coming from and then foster the relationship."* (Sophia, researcher and research user)

4) Partners foster honest and transparent communication

Honest and transparent communication were recognized as core principle for a successful partnership. Participants explained that although difficult, having honest conversations and clear feedback, are important for a partnership to work together effectively. One researcher explained that honesty and transparency are the foundations for supporting meaningful engagement and collaboration:

*"Honesty would be an important one…I think that comes along with transparency about what your expectations are for the project or the amount of time somebody is going to commit to it…and being that like human being and being respectful of one another I think really goes a long way and it really can help you towards building a relationship."* (Isabella, researcher and research user)

## Maintenance

**Build capacity to sustain the network and its partnerships.** A common theme was the need for capacity building among researchers, trainees and research users to sustain the network and its partnerships. In line with the findings described in the first theme, members recognized that the strong leadership and mentorship demonstrated within the CDPP required time and effort and that the sustainability of this level of leadership and mentorship is challenging. With the growth of the network, new leaders and mentors are needed. One research user talked about how current members should step up to take on leadership roles:

*"…an initiative like this [CDPP] can't survive without somebody sustaining it, and I think…people need to step up to continue it."* (Oliver, research user)

Across all interviews, participants highlighted that sustainability of the network and its partnerships requires resources, including 1) human resources, 2) money and time, and 3) training.

Human resources refer to the people within the CDPP network who actively contribute to the development and sustainability of the network. Participants recognized that human resources are an area of capacity building that the CDPP can leverage moving forward. They emphasized the need for dedicated staff or personnel to connect and liaise across research groups, coordinate projects, and serve in roles, such as liaisons or coordinators. Human resources are also essential for fostering and maintaining relationships between researchers and research users (e.g., community organizations, policy makers), and for facilitating meaningful human connection, an aspect that aligns closely with the principled approach outlined in the previous theme. Beyond human resources, dedicated time and money were also identified as critical to sustain the network and partnerships. Both researchers and research users recognized that money for partnership is imperative; without fair compensation, partnership work cannot take place. Financial resources (i.e., money) is needed to compensate research users for their time and to support engagement activities to bring people together. However, participants emphasized that money alone is not always sufficient to ensure successful partnership. Research users also need other types of support, such as protected time to engage in the research partnerships. They explained that money is helpful; however, without the ability to designate time outside of their current responsibilities, engaging in a research partnership is difficult:

*"A lot of our community organizations are already maxed out in terms of our work and so to take on additional workload, a little bit more money to pay one of my staff doesn't really help. Sometimes we need to bring in extra people to either backfill a position or to do the actual knowledge translation work. So, sometimes it's just about, a little bit of extra money and sometimes it's about just providing other types of time and resources."* (Oliver, research user)

As illustrated by Oliver, a flexible and tailored approach is essential for successfully supporting and sustaining partnerships over the long term. Each partnership within the network may need something different, and having open conversations about the type of support needed is crucial for moving forward, which also aligns with the principles of transparent communication, mutual benefit, as discussed in the previous theme.

Finally, training and support to conduct and disseminate research in partnership was identified as important area to focus on. Formal and informal training opportunities for trainees, as well as other researchers and research users, will help to build capacity for research partnerships within the network. Participants also talked about the need to continue training and support in knowledge translation activities. Trainees explained that learning this type of knowledge and skills from other CDPP members will help to prepare for their future career. Trainees talked about how the CDPP facilitated opportunities to learn from researchers and research users in the network:

*"I just think about the training that we have done… [the research lead] has done a lot of education within our lab group…really fortunate that I've been exposed to it."* (Charlotte, trainee)

While human resources, money and time, and training are listed as distinct aspects, they are closely interconnected and mutually reinforcing. For example, human resources require financial support for long-term sustainability. Similarly, money and time are essential to provide training opportunities, which then enhance the capacity of human resources. Training plays a key role in building the skills and knowledge needed for meaningful partnerships, ensuring that members are better equipped to work together. Training also helps maximize the impact of the time and money invested, as members who are trained in partnership may be more likely to engage in meaningful and productive partnerships.

At the network level, all three aspects are necessary for long-term sustainability. At the partnership level, however, the type of support may need to be tailored and flexible, depending on the specific needs of each partnership. Aligning with the previous theme, a principled approach, guided by principles as mutual benefit, transparent communication, and human connection, can help foster meaningful and respectful partnerships.

## Discussion

This study described members' views on the *implementation* of a large, multidisciplinary network over time and explored their experiences of the *implementation* and *maintenance* of the CDPP network and its partnerships. Overall, members reflected positively about how the network functioned and were satisfied with the way the CDPP implemented its plans. In the interviews, participants talked about the strong leadership of the network, the lack of feeling meaningfully connected to the network as a whole, and key principles that guide the success of individual research partnerships. Participants reported that (human) resources and continued leadership are needed to sustain the network and its partnerships on the long-term.

### Transformational leadership and mentorship

A key point of the discussion was the strong leadership and mentorship of CDPP leaders. The importance of strong leadership in complex approaches, such as research partnerships, has been highlighted in previous literature [16,33,36]. In a realist review on mechanisms of knowledge translation of domestic violence research, the authors identified dedicated leadership as one of the mechanisms for the use of research findings in practice [16]. They found that dedicated leadership requires resources in terms of time and money, which aligns with the findings from our study.

The leadership style that our participants described seems to align with transformational leadership theory, which involves leadership behaviours that individually motivate followers to seek challenges and reach their full potential while providing a shared vision for the group [37]. The core tenet of transformational leadership theory is that leaders inspire followers to become leaders in their own right [37]. Participants explained how a CDPP leader, who acted as a 'role model' provided mentorship to facilitate the growth of potential future leaders within the network. This finding resonates with the *idealized influence* (i.e., leaders are 'role models') component of transformational leadership [37]. While participants indicated that they did not feel meaningfully connected to the network as a whole, they mentioned that the network leaders implemented a clear and collective vision for the CDPP network. The survey findings also illustrated that CDPP members consistently reported that CDPP goals were clear. These findings align with *inspirational motivation* (i.e., providing a shared goal and having high expectations). The intellectual *stimulation* (i.e., encouraging members to think critically and creatively to address problems) component resonates with the fact that leaders shared leadership responsibilities, as participants reported the autonomy to develop their own partnerships and leadership opportunities within the network. Finally, leaders discussed and showed interest in the particular roles that each member played in the three research streams, recognizing the difficulty of partnerships within different scientific contexts. This resonates with the *individualized consideration* component (i.e., recognizing individual needs and differences). Aligning with our findings, training leaders to use a transformational leadership approach can help to build capacity for research partnerships. Transformational leadership has been suggested previously as a promising and effective style to make system changes and support research partnerships [33].

Identifying and fostering strong leadership and mentorship to build capacity for research partnerships among different groups (researchers, trainees, research users) requires different strategies. For example, strategies focusing on stimulating individuals' internal motivators may be more effective than focusing on external motivators (e.g., financial compensation). The use of 'role models' has been identified previously as an effective strategy to build capacity for research partnerships within and outside the disability research context [38]. Such strategies require different types of resources (human, time, money, training), in which a tailored approach may be needed to best support individual partnerships within the network. Systemically, funding opportunities, such as the Canadian Social Sciences and Humanities Research Council Partnerships grants, that aim to support networks and partnerships between researchers and research users are needed to continue to build capacity for research partnerships. We encourage funding agencies and organizations from across the world to initiate similar funding opportunities that specifically focus on supporting collaborations between researchers and research users. When establishing new networks and partnerships, we recommend leaders to incorporate transformational leadership behaviours to build capacity for research partnerships among researchers, trainees and research users [33].

## Meaningful partnerships vs meaningful network

Participants indicated that they felt meaningfully connected to their own individual partnerships, but not to the whole CDPP network. The question remains whether it matters if participants had meaningful engagement within their partnerships, but did not feel connected to the broader network. The lack of connection to the network could be explained by the fact that leadership and mentorship activities mainly focused on supporting individual partnerships and collaborations within the team areas. Limited formal, organized activities have been undertaken to facilitate network function as a whole and stimulate collaborations between teams. Participants indicated that collaborations between teams was complicated due to differences in partnership development stages and team operations. These findings align with the commonly reported challenges of doing multidisciplinary research in the areas of participation and disability (e.g., communication challenges, different perspectives). While recognizing the value of different perspectives, the three CDPP teams (sport and exercise, mobility, and employment) decided to continue their partnerships on their own after the grant funding ended. By doing so, separate collaborations and partnerships can be fostered and sustained within each research domain and the continued growth of the networks is expected to be more manageable. A more manageable and connected network is expected to be more impactful in terms of the uptake of research findings. In sum, these findings illustrate the importance of considering the stage of partnership development, the feasibility of doing interdisciplinary research, and the continued growth of the network when building new networks and partnerships.

## Research partnerships: A principle-driven approach

We described four principles to guide the success of research partnerships based on CDPP members' experiences. These four key principles (i.e., valuing research partnership, mutual benefit, human connection, and honest and transparent communication) align with guiding principles for research partnerships or research co-production described in previous literature (e.g., [5,6,31,39,40]). The first principle on valuing research partnership resonates with previous literature describing that partners are internally motivated to conduct their research in partnership [23]. Similarly, Burton et al. (2022) [41] identified that taking a values-driven approach (i.e., being motivated and drive by core beliefs and opinions) is a key feature of people who conduct research in partnership. The second principle on mutual benefit or reciprocity is one that has been consistently reported as a guiding principle or value of different types of research partnership approaches in different areas [5,6,31,39,40]. The principle on human connection or relationship building has been considered as the foundation of research partnerships [33,41]. This relational aspect of working in partnership is also an important category for perceived facilitators (e.g., trust, communication) and barriers (e.g., conflict, feeling of dissatisfaction) to conducting and dissemination research in partnerships [21]. Previous literature [20] indicated the need for building capacity in relational

skills, including effective communication, to promote research partnership approaches, which aligns with the last principle on honest communication. Our findings, again, highlight the principled approach of research partnerships as the guide for success. The fact that participants were able to talk about these key partnership principles may reflect their experiences and knowledge on partnered research reflecting the successful implementation of the network in terms of promoting and supporting research partnerships.

## Implications and lessons learned

To our knowledge, this is the first longitudinal study reporting on the implementation and maintenance of large, multidisciplinary network of partnerships in the area of participation and disability. This study adds to the existing disability literature, by providing an example of how a network of researchers, trainees and research users can be implemented and sustained to promote research partnerships, and ultimately, enhance the uptake of research findings. To illustrate the impact of the CDPP network, over 100 knowledge mobilization resources and products have been published and are available on the CDPP website (www. cdpp.ca), spanning the domains of employment, mobility, and expertise and sport. These resources include blueprints, toolkits, and guides that provide accessible, evidence-informed information for a broad range of audiences. For example, the Blueprint for Building Quality Participation in Sport for Children, Youth, and Adults with a Disability has been widely used by organizations such as the Canadian Paralympic Committee (CPC) to inform inclusive sport programming. CDPP research has also contributed to increased attention on creating effective, equitable, and inclusive workplaces for all individuals, including those with disabilities, by providing employers and organizations with resources to promote inclusive workplaces. The value and continued relevance of CDPP's research in exercise and sport is further demonstrated by the establishment of the CDPP 2.0 network, which includes 29 community partners who continue to benefit from and contribute to this work.

This study adds to the existing partnership literature by providing longitudinal data on members' views on network functioning. While we provide a first step in measuring network function over time, more research is needed to develop reliable and validated methods and approaches to longitudinally monitor and evaluate the impact of a network of research partnerships without overburden participants.

Based on our findings and experiences, we identified three key lessons learned for building and sustaining a multidisciplinary network of research partnerships in the area of participation and disability:

1) **Foster transformational leadership behaviours to build capacity for principle-based research partnerships among researchers, trainees and research users.** This includes mentoring and training early-career researchers, trainees and research users in becoming transformational leaders in research partnerships. Mentoring and training the future generation is needed to successfully sustain the network and its partnerships over time, and has become a key component of the CDPP 2.0 network.

2) **When establishing a multidisciplinary network, consider the stage of partnership development, the feasibility of doing multidisciplinary research, and the potential growth of the network over time.** For example, the CDPP decided to move forward as three separate networks. Continuing with networks within the same research domain may help members to feel connected to both the network and their partnerships.

3) **Formulate shared principles or values to guide collaborations and partnerships within the network**. For example, the IKT Guiding Principles have been developed to support conducting and disseminating spinal cord injury research in partnerships [31]. These principles will also be used to guide and evaluate future partnerships in the CDPP 2.0 network.

## Limitations

This study has some limitations to report. First, the survey responses included only a few responses from research users' perspectives. As such, the disaggregated findings reported in S3 Table should be interpreted with caution. Further, due to

COVID-19 we did not collect survey-data in the year 2020. However, we were still able to yield results over 3 years and captured a unique longitudinal dataset. Second, we have limited information about participants' demographics (e.g., gender, ethnicity). Therefore, we cannot comment on the diversity of our sample, which limits our understanding of the context of our participants' responses. Future studies are encouraged to collect this demographic and background information. Furthermore, the interview data indicated that, in general, participants were very positive about their collaborations and research partnerships. It is possible that CDPP members with positive partnership experiences were more likely to participate in our study compared to those with more negative experiences. As such, our findings should be interpreted with caution. Moreover, interviews were conducted by three different interviewers and then analyzed by a fourth researcher. The change in researchers may have impacted the analyses; however, steps to ensure qualitative rigour were enacted (e.g., critical friends). Another limitation is the limited depth of data collected at the individual project level, as interviews focused on both network-level and project-level experiences and were conducted at a single time point. While participants shared valuable insights into their CDPP projects and partnerships, this design limited our ability to conduct the planned longitudinal multiple case analysis. Future follow-up interviews could be conducted within the context of CDPP 2.0 to enhance our understanding of how project-level partnerships are sustained over time. Finally, we reported on participants' views and experiences regarding the *implementation* and *sustainability* of the network and their partnerships, which targets only two components of the RE-AIM framework. Future studies are needed to identify the extent to which a network of partnerships can contribute to improving translation of research findings (i.e., reach and effectiveness).

## Conclusion

This study provides unique longitudinal insights into the implementation of a multidisciplinary network of research partnerships in the areas of participation and disability. The findings highlighted that building and sustaining a large network of partnerships is challenging and requires effective leadership. We describe lessons learned on how networks of researchers and researcher users may help to build capacity for conducting and disseminating research in partnership, and subsequently may improve the translation of research to practice and policy.

## Supporting information

**S1 Table. Team members' roles and positionality statements.**
(DOCX)

**S2 Table. Our collaborative research activities.**
(DOCX)

**S3 Table. Disaggregated survey findings on implementation component.**
(DOCX)

**S4 Table. COREQ checklist.**
(PDF)

**S1 Text. Interview guide.**
(DOCX)

## Acknowledgments

The authors would like to thank Adrienne Sinden in providing administrative support throughout the project. The authors would like to acknowledge that the majority of this work took place on the unceded territory of the Syilx (Okanagan) Peoples.

## Author contributions

**Conceptualization:** Femke Hoekstra, Kathleen A. Martin Ginis, Veronica Allan, Anita Kothari, Christopher B. McBride.

**Data curation:** Femke Hoekstra, Sarah Lawrason.

**Formal analysis:** Femke Hoekstra, Alanna Shwed, Sarah Lawrason, Veronica Allan, Anita Kothari, Christopher B. McBride, Heather L. Gainforth.

**Funding acquisition:** Kathleen A. Martin Ginis, Christopher B. McBride, Heather L. Gainforth.

**Investigation:** Sarah Lawrason, Heather L. Gainforth.

**Methodology:** Femke Hoekstra, Kathleen A. Martin Ginis, Veronica Allan, Anita Kothari, Christopher B. McBride, Heather L. Gainforth.

**Project administration:** Femke Hoekstra, Sarah Lawrason.

**Resources:** Kathleen A. Martin Ginis, Heather L. Gainforth.

**Software:** Femke Hoekstra, Alanna Shwed.

**Supervision:** Kathleen A. Martin Ginis, Heather L. Gainforth.

**Validation:** Femke Hoekstra.

**Writing – original draft:** Femke Hoekstra, Alanna Shwed, Sarah Lawrason.

**Writing – review & editing:** Femke Hoekstra, Alanna Shwed, Sarah Lawrason, Kathleen A. Martin Ginis, Veronica Allan, Anita Kothari, Christopher B. McBride, Heather L. Gainforth.

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
