## [Decision Letter · Decision Letter 0]

18 Jun 2025

Dear Dr. Hoekstra,

Thank you for submitting your manuscript to PLOS ONE. After careful consideration, we feel that it has merit but does not fully meet PLOS ONE’s publication criteria as it currently stands. Therefore, we invite you to submit a revised version of the manuscript that addresses the points raised during the review process.

**You can find the reviewer's comments below.**

We look forward to receiving your revised manuscript.

Kind regards,

Ivan Sarmiento

Academic Editor

PLOS ONE

Journal Requirements:

This study was supported by a partnership grant from the Social Sciences and Humanities Research Council of Canada (grant no. 895-2013-1021) for the Canadian Disability Participation Project (www.cdpp.ca ).

5. Thank you for uploading your study's underlying data set. Unfortunately, the repository you have noted in your Data Availability statement does not qualify as an acceptable data repository according to PLOS's standards.

6. Please remove all personal information, ensure that the data shared are in accordance with participant consent, and re-upload a fully anonymized data set.

Reviewers' comments:

Reviewer's Responses to Questions

**Comments to the Author**

1. Is the manuscript technically sound, and do the data support the conclusions?

Reviewer #1: Yes

Reviewer #2: Yes

2. Has the statistical analysis been performed appropriately and rigorously?

Reviewer #1: Yes

Reviewer #2: Yes

3. Have the authors made all data underlying the findings in their manuscript fully available?

Reviewer #1: Yes

Reviewer #2: Yes

4. Is the manuscript presented in an intelligible fashion and written in standard English?

Reviewer #1: Yes

Reviewer #2: Yes

Reviewer #1: Thank you for the opportunity to review this manuscript, which explores the implementation and sustainability of the CDPP network through a mixed-methods design. The focus on partnership-based research networks in the field of disability studies is timely and relevant, particularly as collaborative and interdisciplinary approaches continue to grow.

Below are detailed comments aimed at enhancing the clarity, coherence, and overall contribution of the manuscript:

a. The second objective was meant to explore both network-level and partnership-level experiences. However, because of a change in methods, both levels were covered in one interview session. While this change is understandable, the paper does not fully explain how it may have affected the results. This shift might have made it harder to clearly understand partnership-level sustainability, which is an important part of the study. It would be helpful if the authors discussed how this change may have shaped the detail and focus of the findings.

b. The paper introduces common challenges to partnerships, such as time pressures, communication issues, and funding problems. However, these do not appear in the findings. The authors should clarify whether participants did not mention these issues, or if they were raised but left out of the analysis, and explain why.

c. The idea that strong networks can help research influence policy and practice is convincing, but the paper does not include any examples. If there are examples of research from the network being used in practice, it would strengthen the argument to include them. If there were no such cases, the authors might consider adjusting the conclusion to reflect that. Were there any cases where research led to changes in clinical work, community programs, or policies?

d. The paper could benefit from more information on how aspects of the network such as leadership, structure, or ways of engaging people changed over time. This would help readers better understand how the network developed.

e. It would also be helpful to explain how the findings were shared with, or used by, network members or community partners.

f. They say themes were constructed in discussion with co-authors and critical friends but how were the themes derived?

g. Please clarify how consistency and credibility were ensured during the early coding stages.

Reviewer #2: Review of Experiences on the implementation and maintenance of the Canadian Disability Participation Project: a mixed-methods study

The authors present original research that is a component part of an evaluation of the Canadian Disability Participation Project networking using the RE-AIM framework. According to the authors, “More specifically, this study aimed to:

1) describe the implementation of the CDPP network over time;

2) explore members’ experiences and reflections on the implementation and maintenance of the CDPP network and its partnerships.”

My assessment of this mixed methods research is that the design and methods are generally sound but require additional attention in order to ensure coherence throughout the manuscript and generate conclusions that best reflect the data that were generated.

Below is a list of issues that I identified through my review:

Clarity in meaning about Implementation

On page 6, the authors further detail the meanings of the two RE-AIM components that were studied, stating that “Implementation focused on CDPP members’ views on how the network functioned over time and their experiences of being part of the network and its partnerships (lines 107-8).” In their description of the data collection (survey), the authors initially repeat this focus on “views on how the network functioned” and “experiences of being part of” the network. In the lines that follow, the first concept of interest remains consistent while the second becomes “network satisfaction” (p. 10, line 199). This change is not explained, leaving uncertainty about the authors’ focus. I recommend that the authors clarify the second part of the implementation component, either by using consistent descriptions throughout the manuscript or by specifying how experiences and satisfaction are related to one another as part of implementation.

Clarity about network membership

It is unclear to me who was considered to be a member of the network and how big the network was. This uncertainty is particularly relevant for research users who are presented as both “CDPP members” (p. 9, line 179) and researchers’ partners (p. 9, line 184).

Clarity about membership is particularly relevant to determine the response rate, an issue that seems to be of concern to the authors given the limitations identified with respect to the small numbers of survey responses from research users and the possibility of a response bias.

From my read of the authors’ materials, the most promising description of the network make-up and size is the figures representing the CDPP network 2014-2021 that are included in the interview guide. Are these accurate portrayals of the numbers and distribution of the people involved? How do the authors attend to a potentially blurred boundary between a research user being a “community partner” or a “network member”?

I also note that in the text, the authors state, “S3 Text provides additional details on how the CDPP operated, including a visual of the governance structure” (p. 9 lines 174-6). I believe that this is an error as S3 contains the interview guide and I was not able to find the governance structure amid the appendices.

Clarity in the Maintenance (Capacity building) theme

The authors present one theme, “Build capacity to sustain,” as a finding from the Maintenance component of the study. Although not explicitly linked, the theme seemed to be related to and supportive of the first theme, “Strong leadership.”

The theme was presented through a description of the resources required for capacity building, “1) human resources, 2) money and time, and 3) training” (p. 22, line 427). From this list, items 1 and 2 are the least clear to me (I think that training is effectively described at the bottom of page 23). Specifically, what is meant by “human resources”? Are “human resources" created through inputs of money or time as I think is suggested by the quote from Oliver? Is money and time one concept in which money can be converted to time and time to money? If not (as again suggested by Oliver, possibly) what are the conditions that sometimes allow “a little bit of extra money” to buy the time that is needed as compared to those where “a little bit of extra money…doesn’t really help”?

In reading the authors’ description of this theme, I was yearning for clarification about each of the listed items and how they fit together. Most notably, I had the sense that this is not a list of three items (or maybe four), but instead some sort of triad where the items interact to produce the capacity that is necessary for leadership. Of course, my sense of what these concepts are and how they fit together is somewhat speculative – based upon the minimal data and analysis that were presented. A return to the interview transcripts would be necessary to effectively flesh out the participants’ views related to network maintenance.

Should the authors enhance this theme, I would equally suggest that they also review the last part of the Transformational leadership and mentorship sub-section of the Discussion.

**Do you want your identity to be public for this peer review?** For information about this choice, including consent withdrawal, please see our Privacy Policy

Reviewer #1: No

Reviewer #2: **Yes: ** Shaun Cleaver

---

## [Author Response · Author response to Decision Letter 1]

7 Aug 2025

Please see attached document for our detailed responses to the reviewers.

---

## [Decision Letter · Decision Letter 1]

2 Oct 2025

Experiences on the implementation and maintenance of the Canadian Disability Participation Project: a mixed-methods study

PONE-D-24-33799R1

Dear Dr. Hoekstra,

We’re pleased to inform you that your manuscript has been judged scientifically suitable for publication and will be formally accepted for publication once it meets all outstanding technical requirements.

Kind regards,

Ivan Sarmiento

Academic Editor

PLOS ONE

Additional Editor Comments (optional):

Reviewers' comments:

Reviewer's Responses to Questions

**Comments to the Author**

Reviewer #1: All comments have been addressed

2. Is the manuscript technically sound, and do the data support the conclusions?

Reviewer #1: Yes

3. Has the statistical analysis been performed appropriately and rigorously?

Reviewer #1: Yes

4. Have the authors made all data underlying the findings in their manuscript fully available?

Reviewer #1: Yes

5. Is the manuscript presented in an intelligible fashion and written in standard English?

Reviewer #1: Yes

Reviewer #1: The authors of the manuscript have satisfactorily addressed all the comments raised in the previous round

**Do you want your identity to be public for this peer review?** For information about this choice, including consent withdrawal, please see our Privacy Policy

Reviewer #1: **Yes: ** Andres Rojas Cardenas

---

## [Editor Report · Acceptance letter]

PONE-D-24-33799R1

PLOS ONE

Dear Dr. Hoekstra,

I'm pleased to inform you that your manuscript has been deemed suitable for publication in PLOS ONE. Congratulations! Your manuscript is now being handed over to our production team.

Kind regards,

on behalf of

Dr. Ivan Sarmiento

Academic Editor

PLOS ONE